# Free Vibration Analysis of Functionally Graded Porous Cylindrical Panels Reinforced with Graphene Platelets

**DOI:** 10.3390/nano13091441

**Published:** 2023-04-22

**Authors:** Jin-Rae Cho

**Affiliations:** Department of Naval Architecture and Ocean Engineering, Hongik University, Jochiwon, Sejong 30016, Republic of Korea; jrcho@hongik.ac.kr; Tel.: +82-44-860-2546

**Keywords:** GPL-reinforced composite, functionally graded, porous cylindrical panel, free vibration characteristics, 2-D natural element method, MITC3+shell element

## Abstract

The free vibration of functionally graded porous cylindrical shell panels reinforced with graphene platelets (GPLs) was numerically investigated. The free vibration problem was formulated using the first-order shear deformation shell theory in the framework of the 2-D natural element method (NEM). The effective material properties of the GPL-reinforced shell panel were evaluated by employing the Halpin–Tsai model and the rule of mixtures and were modified by considering the porosity distribution. The cylindrical shell surface was transformed into the 2-D planar NEM grid to avoid complex computation, and the concept of the MITC3+shell element was employed to suppress shear locking. The numerical method was validated through benchmark experiments, and the free vibration characteristics of FG-GPLRC porous cylindrical shell panels were investigated. The numerical results are presented for four GPL distribution patterns (FG-U, FG-X, FG-O, and FG-Λ) and three porosity distributions (center- and outer-biased and uniform). The effects of GPL weight, porosity amount, length–thickness and length–radius ratios, and the aspect ratio of the shell panel and boundary condition on the free vibration characteristics are discussed in detail. It is found from the numerical results that the proposed numerical method accurately predicts the natural frequencies of FG-GPLRC porous cylindrical shell panels. Moreover, the free vibration of FG-GPLRC porous cylindrical shell panels is significantly influenced by the distribution pattern as well as the amount of GPLs and the porosity.

## 1. Introduction

Graphene platelets (GPLs) have been widely used as advanced nanofiller materials to improve the physical properties of composites due to their extraordinary physical properties [1,2]. The elastic modulus of GPLs is much higher than that of polymers, so that the structural stiffness of polymer composites can be dramatically increased when only a small amount of graphene sheets are added [3]. Furthermore, it has been reported that graphene provides a higher aspect ratio, larger specific surface area, and lower production costs than carbon nanotubes (CNTs). Moreover, it has been found that the elastic modulus, tensile strength, and fracture toughness of graphene-platelet-reinforced composites (GPLRCs) are higher than those of CNT-reinforced nanocomposites [4]. However, the reinforcement with GPLs is subjected to the limitation of weight fraction owing to the high cost of FGPLs. Owing to this limitation, GPLs are generally blended into the polymer matrix according to the concept of functionally graded materials (FGMs) [5]. FGMs are characterized by the continuous and functional distributions of reinforcement materials through the thickness of composite structures [6]. The physical behavior of GPL-reinforced composites is influenced by this functional distribution pattern, and several purposeful distribution patterns of GPLs have been introduced and investigated: FG-U, FG-V, FG-X, FG-O, and FG-Λ [7,8].

More recently, the GPL-reinforced composite has been developed into a closed-cell porous reinforced foam due to rapidly developed nanotechnology [9,10,11]. As a typical example of porous materials, foams are characterized by high porosity, low density, and large specific surface. Thus, the porous GPL-reinforced composites can synergistically combine the excellent properties of porous foams and GPL nanocomposites [12]. The variation in microstructure and porosity within a porous material can also be purposefully tailored according to the concept of FGMs to enhance the designed material’s performance. These so-called functionally graded porous materials have attracted great attention for the development of next-generation lightweight structures [13]. The porous composite materials reinforced with GPLs in which both internal pores and GPLs are functionally distributed through the structure’s thickness are called functionally graded GPL-reinforced composite (FG-GPLRC) porous structures. In general, the functional distribution patterns are different to each other.

According to the literature survey on FG-GPLRC porous structures, Kitipornchai et al. [14] presented a micromechanics model of FG-GPLRC beams using Timoshenko beam theory and the Ritz method and parametrically investigated the free vibration and buckling behaviors. Gao et al. [15] performed a nonlinear primary resonance analysis of FG-GPLRC porous cylindrical shells under a uniformly distributed harmonic load. Akbas [16] investigated the free vibration and bending deformation of a simply supported functionally graded plate with the porosity effect using the first-order shear deformation plate theory. Barati and Zenkour [17] investigated the postbuckling behavior of geometrically imperfect porous beams reinforced with graphene platelets and resting on a nonlinear hardening foundation. Sahmani et al. [18] presented the size-dependent nonlinear bending of FG-GPLRC porous beams subjected to a uniform distributed load and an axial compressive load using the nonlocal strain gradient theory. Zhou et al. [19] performed a nonlinear buckling analysis of FG-GPLRC porous cylindrical shells under axial compressive load by considering the pre-buckling effect and in-plane constraint. Liu et al. [20] presented an analytical approach for nonlinear static responses and the stability of FG-GPLRC porous arches based on the Euler–Bernoulli hypothesis. Nguyen et al. [21] presented an efficient polygonal finite element method (PFEM) to numerically investigate the static and free vibrations of FG-GPLRC porous plates using Timoshenko’s beam theory. Tao and Dai [22] investigated the postbuckling behavior of sandwich cylindrical shell panels with an FG-GPLRC porous core and two metallic face layers using a general higher-order shear deformation shell theory. Wang and Zhang [23] investigated the thermal buckling and postbuckling behaviors of FG-GPLRC porous beams by considering the temperature-dependent material properties. Wang et al. [24] investigated the forced vibration of FG-GPLRC porous cylindrical microshells subjected to time-dependent distributed impulsive loads. Zhang et al. [25] investigated the thermal buckling of FG-GPLRC porous cylindrical panels by considering the temperature-dependent material properties.

As revealed in the relevant literature survey, most of the studies on FG-GPLRC porous structures have been confined to beam and plate structures by analytical approaches based on the shear deformation theory or by numerical approaches using FEM. The studies on cylindrical structures have been poorly presented and mostly restricted to the buckling behavior. In this context, the purpose of this study was to thoroughly examine the free vibration of FG-CNTRC porous cylindrical shell panels by developing a 2-D effective and locking-free meshfree-based numerical method. The numerical method was developed in the framework of the 2-D planar natural element method (NEM), a previously introduced meshfree method characterized by high smooth Laplace interpolation functions [26,27], in which a geometry transformation and the MITC (mixed-interpolated tensorial components) approach [28] are integrated in order to relax the painstaking manipulation on the shell surface and the troublesome shear locking which occurs in the bending-dominated thin structures [29]. The displacement field of the cylindrical shell is expressed by the first-order shear deformation shell theory, and the physical shell surface and the 2-D rectangular planar NEM grid are correlated through the geometric transformation.

The numerical method was verified through experiments of benchmark examples, and the free vibration responses of FG-GPLRC porous cylindrical shell panels were investigated with respect to the GPL- and porosity-associated parameters, shell geometry dimensions, and boundary conditions. Following the introduction, the FG-GPLRC porous cylindrical shell panel and its displacement, strain and stress fields, and effective material properties are described in Section 2. The natural element approximation of the FG-GPLRC porous cylindrical panel is fully explained in Section 3. The benchmark and parametric numerical experiments are presented in Section 4 with a discussion on the numerical results. The final conclusion is made in Section 5.

## 2. Modeling of FG-GPLRC Porous Cylindrical Shell Panel

Figure 1a shows a cylindrical shell panel reinforced with graphene platelets (GPLs), where a coordinate system $\left(x,y,z\right)$ is introduced on the mid-surface ϖ of the panel using the relation of x=Rθ. The geometric dimensions of the cylindrical panel are represented by radius R, length 𝓁, sub-tended angle $\theta_{0}$, and uniform thickness h. Graphene platelets in this study were distributed according to a specific functionally graded pattern through the thickness. Figure 1b depicts the four patterns adopted for this study, where the GPLs are uniformly dispersed in FG-U, whereas they are rich at the mid-surface in FG-O, at the top surface in FG-X, and at the bottom surface in FG-Λ.

Letting $V_{GPL}\left(z\right)$ and $V_{m}\left(z\right)$ be the volume fractions of GPLs and the underlying matrix material, then both satisfy the physical relation given by
(1)$$V_{GPL}\left(z\right)+V_{m}\left(z\right)=1$$

In which the GPL volume fraction $V_{GPL}\left(z\right)$ is expressed by different functions in terms of the thickness coordinate z and the total GPL volume $V_{GPL}^{*}$ such that
(2)$$V_{GPL}\left(z\right)=\left\{\begin{array}{ll} V_{GPL}^{*}, & \mathrm{FG}\;-\;U\;\\ 2\left(1-2\left|z\right|/h\right)V_{GPL}^{*}, & \mathrm{FG}-O\;\\ 2\left(2\left|z\right|/h\right)V_{GPL}^{*}, & \mathrm{FG}-X\\ \left(1-2z/h\right)V_{GPL}^{*}, & \mathrm{FG}-\Lambda \end{array}\right.$$
for different GPL distribution patterns, where
(3)$$V_{GPL}^{*}=\frac{g_{GPL}}{g_{GPL}+\rho_{GPL}\left(1-g_{GPL}\right)/\rho_{m}}$$
with $g_{GPL}$ being the GPL mass fraction, and $\rho_{GPL}$ and $\rho_{m}$ being the densities of the GPLs and matrix material.

The GPLs are assumed to be uniformly dispersed within the matrix and act as an effective rectangular solid fiber with length $l_{GPL}$, width $w_{GPL}$, and thickness $t_{GPL}$, and the graphene-reinforced composites are modeled as an isotropic material with the effective material properties. The effective elastic modulus $E_{C}$ of GPLRC is estimated by the Halpin–Tsai micromechanical model [30], which gives
(4)$$E_{C}=\frac{3}{8}\cdot \frac{1+\xi_{L}\eta_{L}V_{GPL}}{1-\eta_{L}V_{GPL}}E_{m}+\frac{5}{8}\cdot \frac{1+\xi_{T}\eta_{T}V_{GPL}}{1-\eta_{T}V_{GPL}}E_{m}$$
with
(5)$$\eta_{L}=\frac{E_{GPL}-E_{m}}{E_{GPL}+\xi_{L}E_{m}},\;\eta_{T}=\frac{E_{GPL}-E_{m}}{E_{GPL}+\xi_{T}E_{m}}$$

Here, $E_{GPL}$ and $E_{m}$ denote the elastic moduli of the GPLs and matrix material, and $\xi_{L}$ and $\xi_{T}$ are the geometric parameters given by
(6)$$\xi_{L}=\frac{2l_{GPL}}{t_{GPL}},\;\xi_{T}=\frac{2w_{GPL}}{t_{GPL}}$$

Meanwhile, the effective mass density $\rho_{C}$ and Poisson’s ratio $\nu_{C}$ of the GPLRC are estimated as
(7)$$\rho_{C}=V_{GPL}\rho_{GPL}+V_{m}\rho_{m}$$
(8)$$\nu_{C}=V_{GPL}\nu_{GPL}+V_{m}\nu_{m}$$
according to the linear rule of mixture.

Figure 2 represents three different porosity distributions: center-biased, outer-biased, and uniform, which are expressed as
(9)$$\mathrm{Porosity}\_1:\;\psi \left(z\right)=e_{0}\cdot \cos\left(\frac{\pi z}{h}\right)$$
(10)$$\mathrm{Porosity}\_2:\;\psi \left(z\right)=e_{0}\cdot \left[1-\cos\left(\frac{\pi z}{h}\right)\right]$$
(11)$$\mathrm{Porosity}\_3:\;\psi \left(z\right)=e_{0}$$
with $e_{0}\left(0\leq e_{0}\leq 1\right)$ being the porosity coefficient. The porosity influences the elastic modulus $E_{C}$, shear modulus $G_{C}=E_{C}/2\left(1+\nu_{C}\right)$, and mass density $\rho_{C}$ of the GPLRC. Letting $℘\left(z\right)$ be the effective material properties (i.e., E,G, and ρ) when the porosity is considered, then it is calculated by
(12)$$℘\left(z\right)={℘}_{C}\left(z\right)\cdot \left[1-\psi \left(z\right)\right]$$
from the material properties ${℘}_{C}\left(z\right)$ of the GPLRC. Here, for the effective mass density, the porosity coefficient $e_{0}$ should be modified using the relationship given by
(13)$$\frac{℘\left(z\right)}{{℘}_{C}\left(z\right)}={\left[\frac{\rho \left(z\right)}{\rho_{C}\left(z\right)}\right]}^{2}$$
between the relative mass density and relative elastic property [31]. By letting $e_{m}$ be the porosity parameter for the mass density, it is modified as follows
(14)$$1-e_{m}\cdot \cos\left(\frac{\pi z}{h}\right)=\sqrt{1-e_{0}\cdot \cos\left(\pi z/h\right)}$$
for Porosity 1, for example.

By using the first-order shear deformation shell theory, the displacement field $u={\left\{u_{x},u_{y},u_{z}\right\}}^{T}$ is expressed as
(15)$${\left\{\begin{array}{c} u\\ v\\ w \end{array}\right\}}_{\left(x,y,z\right)}={\left\{\begin{array}{c} u_{0}\\ v_{0}\\ w_{0} \end{array}\right\}}_{\left(x,y\right)}+z\cdot {\left\{\begin{array}{c} \vartheta_{x}\\ \vartheta_{y}\\ 0 \end{array}\right\}}_{\left(x,y\right)}$$
with $d={\left(u_{0},v_{0},w_{0},\vartheta_{x},\vartheta_{y}\right)}^{T}$ being the displacement components at the mid-surface of the shell panel. The strain–displacement relations are expressed as
(16)$$\left\{\begin{array}{c} \varepsilon_{xx}\\ \varepsilon_{yy}\\ 2\varepsilon_{xy} \end{array}\right\}=\varepsilon =\left\{\begin{array}{c} \frac{\partial u_{0}}{\partial x}+\frac{w_{0}}{r}\\ \frac{\partial v_{0}}{\partial y}\\ \frac{\partial v_{0}}{\partial x}+\frac{\partial u_{0}}{\partial y} \end{array}\right\}+z\cdot \left\{\begin{array}{c} \frac{\partial \vartheta_{x}}{\partial x}\\ \frac{\partial \vartheta_{y}}{\partial y}\\ \frac{\partial \vartheta_{y}}{\partial x}+\frac{\partial \vartheta_{x}}{\partial y} \end{array}\right\}=Hd$$
(17)$$\left\{\begin{array}{c} \gamma_{yz}\\ \gamma_{zx} \end{array}\right\}=\gamma =\left\{\begin{array}{c} \vartheta_{y}+\frac{\partial w_{0}}{\partial y}\\ \vartheta_{x}+\frac{\partial w_{0}}{\partial x}-\frac{u_{0}}{r} \end{array}\right\}=H_{s}d$$
with r=R+z≈R. Here, $\left(\varepsilon_{xx},\varepsilon_{yy},\varepsilon_{xy}\right)$ and $\left(\gamma_{yz},\gamma_{zx}\right)$ are in-plane strains and transverse shear strains, and H and $H_{s}$ are the $\left(3\times 5\right)$ and $\left(2\times 5\right)$ partial derivative matrices defined by
(18)$$H=\left[\begin{array}{ccccc} H_{x} & 0 & 1/r & z\cdot H_{x} & 0\\ 0 & H_{y} & 0 & 0 & z\cdot H_{y}\\ H_{y} & H_{x} & 0 & z\cdot H_{y} & z\cdot H_{x} \end{array}\right]$$
(19)$$H_{s}=\left[\begin{array}{ccccc} 0 & 0 & H_{y} & 0 & 1\\ -1/r & 0 & H_{x} & 1 & 0 \end{array}\right]$$
with $H_{x}=\partial /\partial x$ and $H_{y}=\partial /\partial y$. Then, the constitutive relations are expressed as
(20)$$\left\{\begin{array}{c} \sigma_{xx}\\ \sigma_{yy}\\ \sigma_{xy} \end{array}\right\}=\sigma =\frac{E_{C}}{1-\nu_{C}^{2}}\left[\begin{array}{ccc} 1 & \nu_{C} & 0\\ \nu_{C} & 1 & 0\\ 0 & 0 & \left(1-\nu_{C}\right)/2 \end{array}\right]\left\{\begin{array}{c} \varepsilon_{xx}\\ \varepsilon_{yy}\\ 2\varepsilon_{xy} \end{array}\right\}=DHd$$
(21)$$\left\{\begin{array}{c} \tau_{yz}\\ \tau_{zx} \end{array}\right\}=\tau =\left[\begin{array}{cc} G_{C} & 0\\ 0 & G_{C} \end{array}\right]\left\{\begin{array}{c} \gamma_{yz}\\ \gamma_{zx} \end{array}\right\}=D_{s}H_{s}d$$
using the in-plane stresses $\left(\sigma_{xx},\sigma_{yy},\sigma_{xy}\right)$ and the transverse shear stresses $\left(\tau_{yz},\tau_{zx}\right)$.

## 3. Analysis of Free Vibration Using 2-D NEM

For the free vibration analysis of the FG-GPLRC cylindrical shell panel by 2-D NEM, the mid-surface ϖ of the shell panel is discretized into a finite number of nodes and Delaunay triangles, as depicted in Figure 3. Then, the approximate displacement $u^{h}\left(x,y,z\right)$ is expressed as
(22)$${\left\{\begin{array}{c} u^{h}\\ v^{h}\\ w^{h} \end{array}\right\}}_{\left(x,y,z\right)}=\sum_{J=1}^{N}{\left\{\begin{array}{c} u_{0}\\ v_{0}\\ w_{0} \end{array}\right\}}_{J}\psi_{J}\left(x,y\right)+\sum_{J=1}^{N}z\cdot {\left\{\begin{array}{c} \vartheta_{x}\\ \vartheta_{y}\\ 0 \end{array}\right\}}_{J}\psi_{J}\left(x,y\right)$$
using Laplace interpolation functions $\psi_{J}\left(x,y\right)$ [26,32] and the nodal vector $d_{J}={\left(u_{0},v_{0},w_{0},\vartheta_{x},\vartheta_{y}\right)}_{J}^{T}$ of displacement components, where the subscript J indicates the J-th node within the NEM grid ${ℑ}_{C}$ composed of N nodes.

The definition of the Laplace interpolation function and its manipulation on the cylindrical surface are complex and painstaking. To relax this difficulty, a geometry transformation $T_{C}^{}$ is introduced to correlate the physical NEM grid ${ℑ}_{C}=\left[0,R\theta_{0}\right]\times \left[0,𝓁\right]$ on the cylindrical surface and the computational NEM grid ${ℑ}_{R}=\left[0,\theta_{0}\right]\times \left[0,𝓁\right]$ on the rectangular plane with coordinates $\zeta_{1}$ and $\zeta_{2}$:(23)$$T_{C}^{}:\;\left(\zeta_{1},\zeta_{2}\right)\in {ℑ}_{R}\;\to \;\left(x,y\right)\in {ℑ}_{C}$$.

Then, Laplace interpolation functions $\psi_{J}\left(x,s\right)$ are mapped to $\varphi_{J}\left(\zeta_{1},\zeta_{2}\right)$, and the relations of $x=R\cdot \zeta_{1}$ and $y=\;\zeta_{2}$ lead to the inverse Jacobi matrix $J^{-1}$ given by
(24)$$J^{-1}=\left[\begin{array}{cc} \partial \zeta_{1}/\partial x & \partial \zeta_{1}/\partial y\\ \partial \zeta_{2}/\partial x & \partial \zeta_{2}/\partial y \end{array}\right]=\left[\begin{array}{cc} 1/R & 0\\ 0 & 1 \end{array}\right]$$

The partial derivatives $H_{x}$ and $H_{y}$ in Equations (18) and (19) on the cylindrical surface are switched to
(25)$$\frac{\partial }{\partial x}=H_{x}=\frac{1}{R}\frac{\partial }{\partial \zeta_{1}}=\frac{1}{R}H_{1},\;\frac{\partial }{\partial y}=H_{y}=\frac{\partial }{\partial \zeta_{2}}=H_{2}$$
on the rectangular plane according to the chain rule.

Introducing Equation (25) into Equations (18) and (19) results in $\hat{H}$ and ${\hat{H}}_{s}$ in which $H_{x}$ and $H_{y}$ are replaced with $H_{1}$ and $H_{2}$:(26)$$T_{C}^{-1}:\;H,H_{s}\;\to \;\hat{H},{\hat{H}}_{s}$$

Then, the NEM approximations of the in-plane strains ε in Equation (16) and the transverse shear strains γ in Equation (17) lead to
(27)$$\varepsilon^{h}=\sum_{J=1}^{N}\hat{H}\phi_{J}d_{J}=\sum_{J=1}^{N}B^{J}d_{J},\;\gamma^{h}=\sum_{J=1}^{N}{\hat{H}}_{s}\phi_{J}d_{J}=\sum_{J=1}^{N}B_{s}^{J}d_{J}$$

The direct approximation (20) of transverse shear strain γ using the standard $C^{0}-$ interpolation functions can frequently suffer from a big approximation error caused by the shear locking [29,33,34]. To ensure numerical accuracy, the transverse shear strains are indirectly interpolated by adopting the three-node triangular MITC3+shell finite element represented in Figure 4 [28] with coordinates ξ and η. Each triangular element $\varpi_{e}$ in the physical NEM grid ${ℑ}_{C}$ shown in Figure 3 is mapped to its master triangular element $\hat{\varpi }$. Letting $N_{K}\left(\xi ,\eta \right)$ be the linear triangular FE shape functions [35], the approximate displacement field $u^{h}\left(x,y,z\right)$ in Equation (22) is re-interpolated using the element-wise nodal vectors $d_{K}^{e}={\left(u_{0}^{e},v_{0}^{e},w_{0}^{e},\vartheta_{y}^{e},\vartheta_{y}^{e}\right)}_{K}^{T}$ approximated by 2-D NEM.

Then, referring to Figure 4, the element-wise transverse shear strains $\hat{\gamma }$ are interpolated as:(28)$${\hat{\gamma }}_{xz}^{e}=\frac{2}{3}\left[\gamma_{xz}^{\left(B\right)}-\frac{1}{2}\gamma_{yz}^{\left(B\right)}\right]+\frac{1}{2}\left[\gamma_{xz}^{\left(C\right)}+\gamma_{yz}^{\left(C\right)}\right]+\frac{\hat{c}}{3}\left(3\eta -1\right)$$
(29)$${\hat{\gamma }}_{yz}^{e}=\frac{2}{3}\left[\gamma_{yz}^{\left(A\right)}-\frac{1}{2}\gamma_{xz}^{\left(A\right)}\right]+\frac{1}{2}\left[\gamma_{yz}^{\left(C\right)}+\gamma_{xz}^{\left(C\right)}\right]+\frac{\hat{c}}{3}\left(1-3\xi \right)$$
using the transverse shear strains at the tying points A, B, C, and D within the three-node triangular master element and $\hat{c}=\gamma_{xz}^{\left(F\right)}-\gamma_{xz}^{\left(D\right)}+\gamma_{yz}^{\left(E\right)}-\gamma_{yz}^{\left(F\right)}$. The analytic derivation of Equations (28) and (29) using Equation (17), the FE re-interpolation, and the chain rule between two coordinates $\left(x,y\right)$ and $\left(\xi ,\eta \right)$ leads to ${\hat{\gamma }}_{}^{e}={\hat{B}}_{e}^{}d_{e}$ with the $\left(2\times 15\right)$ matrices ${\hat{B}}_{e}^{}$ in function of ξ,η,z and R and the $\left(15\times 1\right)$ element-wise nodal vectors $d_{e}=\left\{d_{1}^{e},d_{2}^{e},d_{3}^{e}\right\}$.

Meanwhile, the standard Galerkin weak form for the free vibration analysis of FG-GPLRC cylindrical shell panels can be derived from the dynamic form of the energy principle [36]
(30)$$\int_{-h/2}^{h/2}\int_{\varpi }\left[{\left(\delta \varepsilon \right)}^{T}D\varepsilon +{\left(\delta \gamma \right)}^{T}D_{s}\gamma \right]d\varpi dz+\int_{-h/2}^{h/2}\int_{\varpi }{\left(\delta d\right)}^{T}m\ddot{d}\;d\varpi \;dz=0$$

Here, m is a $\left(5\times 5\right)$ symmetric matrix defined by:(31)$$m=\rho \left[\begin{array}{cc} I & m_{1}^{T}\\ m_{1} & m_{2} \end{array}\right],\;m_{1}=\left[\begin{array}{ccc} z & 0 & 0\\ 0 & z & 0 \end{array}\right]$$
with the $\left(3\times 3\right)$ identity matrix I and $m_{2}=diag\left(z^{2},z^{2}\right)$. Assuming the harmonic motion $d=\bar{d}\cdot e^{j\omega t}$ and plugging Equations (27)–(29) into Equation (30), together with the constitutive relations (20) and (21), one can derive the modal equation given by:(32)$$\left[\left(K_{\sigma }+\sum_{e=1}^{M}K_{s}^{e}\right)-\omega^{2}M\right]\;\bar{d}=0$$
to solve the natural frequencies ${\left\{\omega_{I}\right\}}_{I=1}^{N}$ and the natural modes ${\left\{{\bar{d}}_{I}\right\}}_{I=1}^{N}$ of the cylindrical panel which is discretized into M Delaunay triangles. Here, two stiffness matrices and the mass matrix are defined by:(33)$$K_{\sigma }=\int_{-h/2}^{h/2}\int_{\varpi }B_{}^{T}DB\;d\varpi dz$$
(34)$$K_{s}^{e}=\int_{-h/2}^{h/2}\int_{\varpi_{e}}{\hat{B}}_{e}^{T}{\hat{D}}_{s}{\hat{B}}_{e}\;d\varpi dz$$
(35)$$M=\int_{-h/2}^{h/2}\int_{\varpi }\Phi^{T}m\Phi \;d\varpi dz$$
where $\bar{d}=\left[d_{1},d_{2},\cdot \cdot \cdot ,d_{N}\right]$, $B=\left[B_{1},B_{2},\cdot \cdot \cdot ,B_{N}\right]$, $\Phi =\left[\Phi_{1},\Phi_{2},\cdot \cdot \cdot ,\Phi_{N}\right]$ with $\Phi_{J}=diag\left[\phi_{J},\phi_{J},\phi_{J},\phi_{J},\phi_{J}\right]$, the material constant matrix D defined in Equation (20), and ${\hat{D}}_{s}$ given by
(36)$${\hat{D}}_{s}=\frac{\beta \cdot \kappa }{1+\alpha \cdot {\left(L_{e}/h\right)}^{2}}\left[\begin{array}{cc} G_{C} & 0\\ 0 & G_{C} \end{array}\right]$$
with the shear correction factor κ=5/6, the longest side length $L_{e}$ of the Delaunay triangles, and a positive constant $\alpha \left(\alpha >0\right)$ called the shear stabilization parameter [34,37]. The value of α is chosen through the preliminary experiment, and this modification of the shear modulus matrix was proposed to stabilize the MITC3 element. Meanwhile, β is the porosity stabilization parameter which is dependent on the porosity distribution pattern.

## 4. Results and Discussion

A Fortran program was coded according to the numerical formulae presented in Section 3 and integrated into the previously developed NEM program [38] which was developed for plate-like structures. The numerical integration of stiffness and mass matrices in Equations (33)–(35) was carried out triangle by triangle using seven Gauss integration points for $K_{\sigma }$ and M and one Gauss point for $K_{s}^{e}$. Referring to Figure 3, uniform 11×11 NEM grids were used for the numerical experiments, and a total of 15 modes were extracted using the Lanczos transformation and Jacobi iteration methods, unless otherwise stated. Meanwhile, three types of boundary conditions, simply supported (S), clamped (c), and free, were considered, where S and C were implemented as
(37)$$S:\;v_{0}=w_{0}=\vartheta_{y}=0$$
(38)$$C:\;u_{0}=v_{0}=w_{0}=\vartheta_{x}=\vartheta_{y}=0$$

The present method was compared with the other methods with two isotropic, one FG-GPLRC, and one FG-GPLRC porous cylindrical panels.

The first example is a clamped isotropic cylindrical panel with the geometry dimensions given by R=𝓁=0.762 m, s=0.1016 m, h=0.00033 m. The elastic modulus E, Poisson’s ratio ν, and density ρ are 68.948 GPa, 0.33, and $2711.5\;kg/m^{3}$, respectively. The maximum relative differences of the numerical results given in Table 1 are 5.819% compared with the experimental data [39] and 1.394% and 1.097% compared with references [40,41], respectively. Thus, it has been verified that the present method is in good agreement with the three reference results.

The second example is the simply supported isotropic cylindrical panels with four different aspect ratios s/𝓁. The relative geometry dimensions are 𝓁/R=0.1,𝓁/h=10, the Poisson’s ratio is ν=0.3, and the fundamental frequencies are calibrated as ${\hat{\omega }}_{1}=\omega_{1}𝓁\sqrt{\rho \left(1-\nu^{2}\right)/E}$. Yang and Shen [41] and Kobayashi and Leissa [42] in Table 2 employed the higher-order and the first-order shear deformation shell theories, while Chen and Chao [42] adopted the 3-D elasticity theory. The maximum relative differences are 0.882% compared with reference [40] and 1.706% and 2.069% compared with references [42] and [43], respectively. It has been confirmed again that the present method is in excellent agreement with the existing reference solutions for different aspect ratios with the maximum relative difference equal to 2.069%.

The third example is the functionally non-porous cylindrical panels reinforced with graphene platelets with different ratios of a/h and R/a for two different boundary conditions, SSSS and CCCC, where the combined four capital letters indicate a set of boundary conditions specified for the four sides ①,②,③, and ④ of the cylindrical panel, as show in Figure 3. The shell radius R and a/b are 10 m and 1.0, and the GPL weight fraction $g_{GPL}$ is set to 1.0%. Epoxy is taken as the polymer matrix and its material properties are $E_{m}=3.0\;GPa,v_{m}=0.34$, and $\rho_{m}=1200\;kg/m^{3}$. Referring to Yasmin and Daniel [44] and Rafiee et al. [4], the geometry dimensions of GPLs are $l_{GPL}=2.5\;\mu m,$
$t_{GPL}=1.5\;nm$, and $w_{GPL}=1.5\;\mu m$ while the material properties are $E_{GPL}=1.01\;TPa,v_{GPL}=0.186$, and $\rho_{GPL}=1060\;kg/m^{3}$. The fundamental frequencies are calibrated as ${\hat{\omega }}_{1}=\omega_{1}{𝓁}^{2}/h\sqrt{\rho_{m}/E_{m}}$, and the computed normalized fundamental frequencies are compared in Table 3 with the numerical results obtained by Van Do and Lee [5] using the isogeometric analysis (IGA) method.

It can be seen in Table 3 that the fundamental frequencies obtained by the present method are as a whole smaller than those obtained by the IGA method, except for several exceptional cases. Regarding the GPL distribution pattern, the relative differences between the present method and the IGA method are shown to be relatively higher at FG-O. Meanwhile, the dependence of the relative difference in ${\hat{\omega }}_{1}$ on the 𝓁/h, R/𝓁 and the boundary condition is not apparent. The maximum relative difference in ${\hat{\omega }}_{1}$ between the two methods is found to be 4.213% at the simply supported FG-O with 𝓁/h=50 and R/𝓁=50. Thus, it has been verified that the present method accurately predicts the fundamental frequencies of FG-GPLRC cylindrical panels for various GPL distribution patterns, geometry dimensions, and boundary conditions.

Next, the normalized fundamental frequency ${\hat{\omega }}_{1}$ of non-porous FG-GPLRC cylindrical panels was parametrically investigated. Figure 5a,b represent the variations in ${\hat{\omega }}_{1}$ with respect to the GPL mass fraction $g_{GPL}$ for different GPL distribution patterns and boundary conditions. It is observed that ${\hat{\omega }}_{1}$ uniformly increases with the increasing value of $g_{GPL}$, regardless of the GPL distribution pattern and the boundary condition. This is because the panel stiffness increase due to the increase in the GPL amount is greater than the panel mass increase. Moreover, the normalized fundamental frequency is remarkably influenced by the GPL distribution pattern and the boundary condition. The order of the magnitude of ${\hat{\omega }}_{1}$ among the four GPL distribution patterns is FG-X > FG-U > FG-Λ> FG-O. This is because the thickness-wise GPL distribution pattern strongly affects the structural stiffness such that the structural stiffness increases as the GPL distribution becomes biased to the top and bottom surfaces of the panel. Regarding the effect of the boundary condition, the order of the magnitude of ${\hat{\omega }}_{1}$ is CCCC > CSCS > CFCF > SSSS, which is consistent with the order of the strength of the boundary constraint.

Figure 6a represents the effect of the length–thickness ratio 𝓁/h on the normalized fundamental frequency, where the panel length 𝓁 is kept unchanged at 1.0 m. It is seen that ${\hat{\omega }}_{1}$ increases in proportion to 𝓁/h, but this owes entirely to the calibration with ${𝓁}^{2}/h$. It was found from the numerical data that the non-calibrated absolute fundamental frequency $\omega_{1}$ decreases with the increasing value of 𝓁/h because the panel stiffness remarkably decreases as the panel becomes thinner. Figure 6b shows the effect of the radius–length R/𝓁 on the normalized fundamental frequency, where the panel length 𝓁 is also kept unchanged at 1.0 m. It is seen that the normalized fundamental frequency increases in reverse proportion to the shell radius R, because the structural stiffness increases while the total mass decreases as the shell radius becomes smaller.

Figure 7a shows the variation in ${\hat{\omega }}_{1}$ with respect to the length–thickness ratio 𝓁/h for different values of $g_{GPL}$. It is seen that ${\hat{\omega }}_{1}$ uniformly increases in proportion to 𝓁/h, and the increase in the slopes is almost the same regardless of $g_{GPL}$. The explanation for why ${\hat{\omega }}_{1}$ increases with the increasing value of 𝓁/h is the same as for Figure 6a. Figure 7b shows the variation in ${\hat{\omega }}_{1}$ with the aspect ratio of the shell panel, where the shell length is kept unchanged at 1.0 m. It is observed that ${\hat{\omega }}_{1}$ dramatically increases in proportion to the value of 𝓁/h, because the decrease in shell width b dramatically increases the panel stiffness but reduces the panel weight. Similar to the length–thickness ratio, the increase in the slope is almost insensitive to the GPL mass fraction $g_{GPL}$.

Next, the free vibration of the FG-GPLRC porous cylindrical panel was investigated by considering three porosity distributions shown in Figure 2. Table 4 compares the normalized fundamental frequencies of the simply supported FG-GPLRC porous cylindrical panel with the reference solutions of Zhou et al. [45]. The GPL distribution pattern is FG-U, and the geometry dimensions are given in the table caption. The reference solutions were obtained by employing Reddy’s third-order shear deformation theory. The fundamental frequencies were normalized as ${\hat{\omega }}_{1}=\omega_{1}R\sqrt{\rho_{m}/E_{m}}$, and the porosity stabilization parameter β included in Equation (36) was determined through the preliminary experiment: β=1 for Porosity_1, $\beta =1/{\left(1+10e_{0}^{2}\right)}^{2}$ for Porosity_2, and $\beta =1/{\left(1-e_{0}^{2}\right)}^{2}$ for Porosity_3. When compared with the reference solutions, the present results are higher at Porosity_1 but lower at Porosity_2 and 3. The maximum relative difference equal to 1.08% occurred at $g_{GPL}=1.0$ and $e_{0}=0.3$ for Porosity_3. Thus, the comparison verifies that the present method accurately predicts the fundamental frequency of FG-GPLRC porous cylindrical panels.

Figure 8a represents the variation in ${\hat{\omega }}_{1}$ with respect to the porosity parameter $e_{0}$, where the ${\hat{\omega }}_{1}$ uniformly decreases in proportion to $e_{0}$. This is because the decrease in panel stiffness owing to the increase in the porosity amount is greater than the decrease in the panel mass. Meanwhile, the decreasing trend is affected by the porosity distribution such that the decrease in the slope is highest at Porosity_2 and lowest at Porosity_1. This is because the decrease in the structural stiffness becomes larger as the porosity distribution becomes biased to the top and bottom surfaces of the panel. Figure 8b represents the effect of the GPL distribution pattern on the variation in ${\hat{\omega }}_{1}$ with respect to the porosity parameter $e_{0}$. It is found that the GPL distribution pattern affects the order of the magnitude of ${\hat{\omega }}_{1}$, but its effect on the decrease in the slope of ${\hat{\omega }}_{1}$ with respect to the porosity parameter $e_{0}$ is not remarkable. This trend is different from the effect of the GPL distribution pattern on the variation in ${\hat{\omega }}_{1}$ with respect to the GPL mass fraction shown in Figure 5a. This is because the decrease in the slope itself of the panel stiffness and the panel mass along the porosity parameter $e_{0}$ is almost insensitive to the GPL distribution pattern.

Figure 9a represents the effect of GPL mass fraction $g_{GPL}$ on the decrease in ${\hat{\omega }}_{1}$ with respect to the porosity parameter $e_{0}$. It is seen that the $g_{GPL}$ affects the magnitude of ${\hat{\omega }}_{1}$, but its effect on the decrease in the slope of ${\hat{\omega }}_{1}$ along the porosity parameter is not significant. This trend is also seen in Figure 9b which represents the effect of the boundary condition on the decrease in ${\hat{\omega }}_{1}$ with the porosity parameter $e_{0}$. It is found that the magnitude of ${\hat{\omega }}_{1}$ is apparently influenced by the boundary condition, but the dependence of its decreasing slope with respect to the porosity parameter on the boundary condition is not shown to be remarkable. This is because the decrease in the slope itself of the panel stiffness and the panel mass with respect to $e_{0}$ is not remarkably influenced by the GPL mass fraction $g_{GPL}$ and the boundary condition.

Figure 10a represents the effect of porosity distribution on the increase in ${\hat{\omega }}_{1}$ with respect to the GPL mass fraction $g_{GPL}$. It is found that the order of the magnitude of ${\hat{\omega }}_{1}$ is apparently affected by the porosity distribution, but the increase in the slope of ${\hat{\omega }}_{1}$ with the GPL mass fraction is not influenced by the porosity distribution. This trend is different from the effect of porosity distribution on the decrease in ${\hat{\omega }}_{1}$ with respect to the porosity parameter shown in Figure 8a. Meanwhile, Figure 10b represents the variation in ${\hat{\omega }}_{1}$ with respect to the GPL mass fraction $g_{GPL}$ for different porosity parameters. It is observed that the porosity parameter significantly affects the magnitude of ${\hat{\omega }}_{1}$, but its effect on the increase in the slope of ${\hat{\omega }}_{1}$ with respect to $g_{GPL}$ is not shown to be remarkable. This is because the increase in the slope itself of the panel stiffness and the panel mass with respect to $g_{GPL}$ is not remarkably influenced by the GPL distribution and the porosity distribution.


## 5. Conclusions

This paper presents a free vibration analysis of FG-GPLRC porous cylindrical shell panels using a NEM-based 2-D numerical method. In the framework of 2-D planar NEM, the numerical method was developed by integrating a geometry transformation between the shell surface and the 2-D planar NEM grid and the MITC3+shell element. Benchmark and parametric experiments were carried out to validate the proposed numerical method and to thoroughly investigate the free vibration characteristics of FG-GPLRC porous cylindrical panels with respect to the associated parameters. The numerical results reveal the following main observations:
The numerical method accurately analyzes the free vibration of FG-GPLRC porous cylindrical shell panels, without causing shear locking, with the maximum relative difference of 4.213% even for coarse and 2-D planar NEM grids.The normalized natural frequency ${\hat{\omega }}_{1}$ uniformly increases in proportion to the GPL mass fraction $g_{GPL}$ while it uniformly decreases with the increasing value of porosity parameter $e_{0}$, and it uniformly increases in proportion to the length–thickness ratio 𝓁/h, the length–radius ratio 𝓁/R, and the aspect ratio 𝓁/b of the shell panel.The distribution patterns of both the GPL and porosity significantly affect the variations in ${\hat{\omega }}_{1}$ with respect to the values of $g_{GPL}$ and $e_{0}$ such that the order of the magnitude of ${\hat{\omega }}_{1}$ among the four GPL distribution patterns is FG-X > FG-U > FG-Λ> FG-O while that among the three porosity distributions is Porosity_1 > Porosity_3 > Porosity_2.The increase in the slope of ${\hat{\omega }}_{1}$ with respect to the GPL mass fraction is influenced by the GPL distribution pattern, 𝓁/h, 𝓁/R, and 𝓁/b, but it is independent of the magnitude and distribution of the porosity. Meanwhile, the decrease in the slope of ${\hat{\omega }}_{1}$ with respect to the porosity parameter is influenced by the porosity distribution, but it is independent of the mass fraction and distribution of the GPL and the boundary condition.


## Figures and Tables

**Figure 1 nanomaterials-13-01441-f001:**
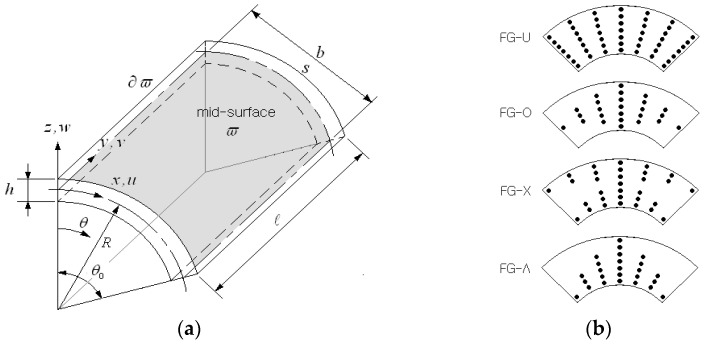
FG-GPLRC cylindrical shell panel: (**a**) geometry and dimensions; (**b**) functionally graded distributions of GPLs.

**Figure 2 nanomaterials-13-01441-f002:**
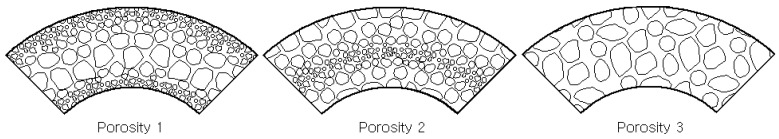
Three different porosity distributions (1: center-biased, 2: outer-biased, 3: uniform).

**Figure 3 nanomaterials-13-01441-f003:**
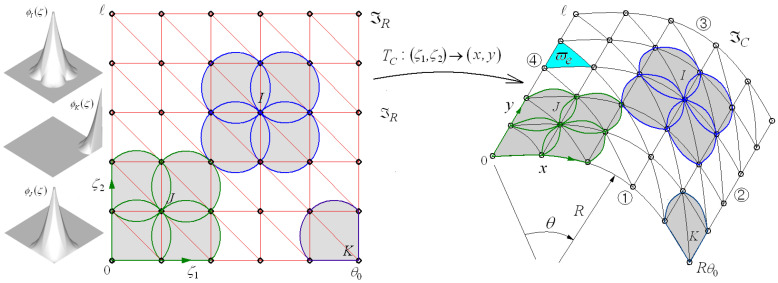
Laplace interpolation functions $\phi_{J}\left(\zeta_{1},\zeta_{2}\right)$ defined on the rectangular plane and their transformation to $\psi_{J}\left(x,y\right)$ on the cylindrical surface.

**Figure 4 nanomaterials-13-01441-f004:**
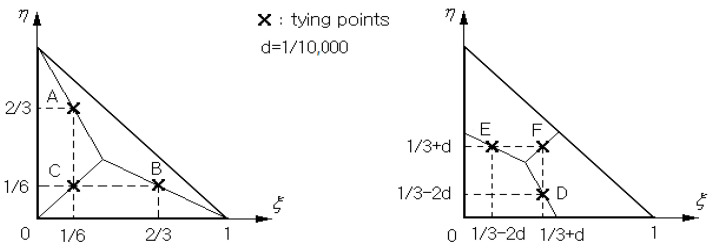
The tying points in the three-node triangular master element $\hat{\varpi }$ for interpolating the transverse shear strains $\hat{\gamma }$.

**Figure 5 nanomaterials-13-01441-f005:**
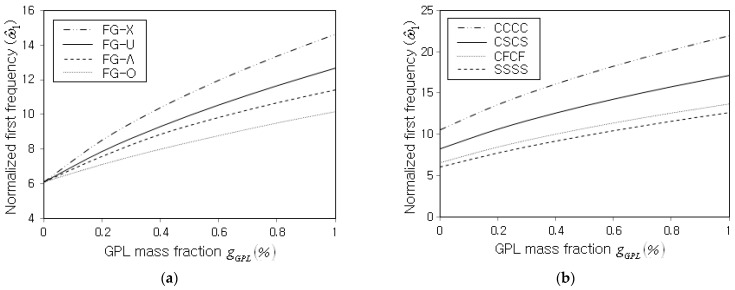
Normalized fundamental frequency with respect to the GPL mass fraction $g_{GPL}$ (R=10 m, R/𝓁=10, 𝓁/b=1.0, 𝓁/h=20): (**a**) effect of the GPL distribution pattern (SSSS); (**b**) effect of the boundary condition (FG-U).

**Figure 6 nanomaterials-13-01441-f006:**
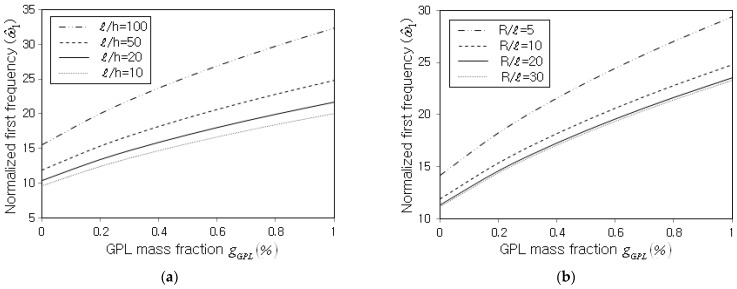
Normalized fundamental frequency with respect to the GPL mass fraction $g_{GPL}$ (CCCC, FG-U): (**a**) effect of the length–thickness ratio (R/𝓁=10); (**b**) effect of the radius–length ratio (𝓁/h=50).

**Figure 7 nanomaterials-13-01441-f007:**
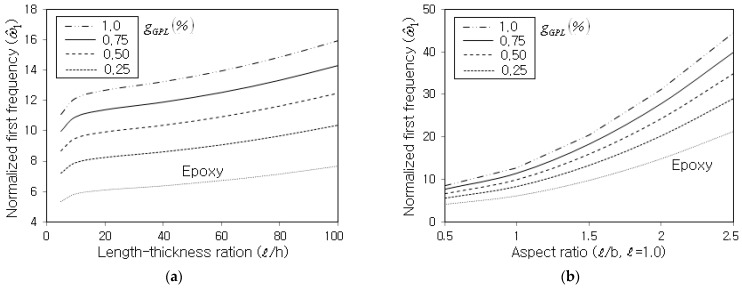
Normalized fundamental frequency (SSSS, FG-U, R=10 m, R/𝓁=10) with respect to: (**a**) the length–thickness ratio (𝓁/b=1.0) and (**b**) the aspect ratio (𝓁/h=20).

**Figure 8 nanomaterials-13-01441-f008:**
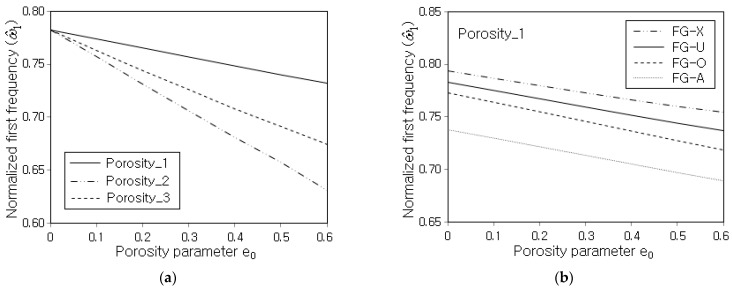
Normalized fundamental frequency with respect to the porosity parameter (SSSS, $g_{GPL}=0.5\%$): (**a**) for different porosity distributions (FG-U) and (**b**) for different GPL distributions (Porosity_1).

**Figure 9 nanomaterials-13-01441-f009:**
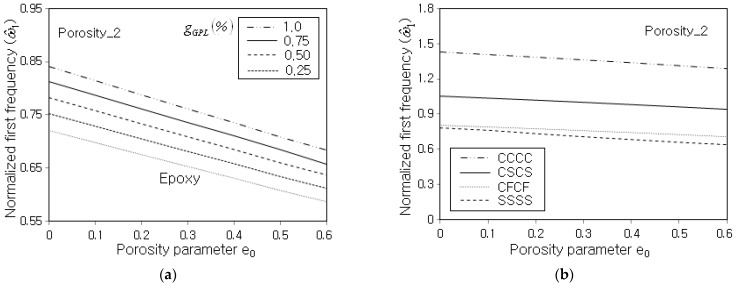
Normalized fundamental frequency with respect to the porosity parameter (Porosity_2, FG-U): (**a**) for different GPL mass fractions (SSSS) and (**b**) for different boundary conditions ($g_{GPL}=0.5\%$).

**Figure 10 nanomaterials-13-01441-f010:**
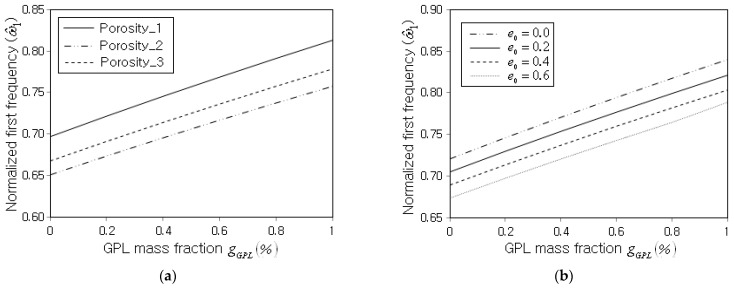
Normalized fundamental frequency with respect to the GPL mass fraction (FG-U, SSSS): (**a**) for different porosity distributions ($e_{0}=0.3$) and (**b**) for different porosity parameters (Porosity_1).

**Table 1 nanomaterials-13-01441-t001:** Comparison of four lowest natural frequencies $\left(Hz\right)$ of the clamped isotropic cylindrical panel.

Mode	Experimental Deb Nath [39]	Numerical		
		Au and Cheung [40]	Yang and Sheng [41]	Present
1	814	869	871	873
2	940	957	961	948
3	1260	1287	1280	1292
4	1306	1363	1367	1382

**Table 2 nanomaterials-13-01441-t002:** Comparison of the normalized fundamental frequencies ${\hat{\omega }}_{1}$ of the simply supported isotropic cylindrical panels.

s/𝓁	Yang and Shen [41]	Kobayashi and Leissa [42]	Chen and Chao [43]	Present
0.5	1.31597	1.3360	1.31742	1.31321
1.0	0.55136	0.5563	0.55049	0.55184
1.5	0.40266	0.4044	0.39987	0.40361
2.0	0.35019	0.3505	0.34612	0.35328

**Table 3 nanomaterials-13-01441-t003:** Comparison of the normalized fundamental frequencies ${\hat{\omega }}_{1}$ of functionally graded porous cylindrical panels reinforced with graphene platelets ($R=10\;m,\;𝓁/b=1.0,\;g_{GPL}=1.0\%$).

Method	𝓁/h	R/𝓁	GPL Distribution Pattern				
			Epoxy	FG-U	FG-O	FG-X	FG-Λ
IGA [5]	20 (SSSS)	10	6.0826	12.6556	10.1648	14.6685	11.4098
		50	6.0057	12.4953	9.9625	14.5317	11.2364
	20 (CCCC)	10	10.8810	22.6400	18.3299	25.8854	20.4855
		50	10.7393	22.3451	17.9653	25.6269	20.1641
	50 (SSSS)	10	6.5434	13.6153	11.2729	15.6009	12.4379
		50	6.0705	12.6301	10.0583	14.7500	11.3555
	50 (CCCC)	10	11.8649	24.6863	20.4352	28.2406	22.5568
		50	11.0295	22.9478	18.3005	26.7320	20.6453
Present	20 (SSSS)	10	6.0776	12.6431	10.4697	14.4563	11.4551
		50	5.9217	12.3178	10.1885	14.0861	11.4991
	20 (CCCC)	10	10.6859	22.2309	18.0054	25.6567	20.3608
		50	10.5473	21.9423	17.6483	25.4070	20.1356
	50 (SSSS)	10	6.5038	13.5306	11.5976	15.1787	11.9887
		50	6.0528	12.5910	10.4821	14.3451	11.6452
	50 (CCCC)	10	11.9282	24.8170	20.9011	28.1536	22.8862
		50	11.1209	23.1368	18.8826	26.6819	21.3045

**Table 4 nanomaterials-13-01441-t004:** Comparison of normalized fundamental frequencies of functionally graded porous cylindrical panels reinforced with graphene platelets (SSSS, FG-U, 𝓁/h=R/h=10, h=0.01 m, α=π/3.

Method	$$g_{GPL}\left(\%\right)$$	$$e_{0}$$	Porosity Distribution Pattern		
			Porosity_1	Porosity_2	Porosity_3
Zhou et al. [19]	0	0.3	0.6904	0.6494	0.6740
		0.6	0.6701	0.5839	0.6272
	0.5	0.3	0.7505	0.7060	0.7327
		0.6	0.7284	0.6347	0.6818
	1.0	0.3	0.8062	0.7583	0.7870
		0.6	0.7824	0.6818	0.7323
Present	0	0.3	0.6959	0.6497	0.6676
		0.6	0.6718	0.5834	0.6229
	0.5	0.3	0.7558	0.7057	0.7252
		0.6	0.7296	0.6339	0.6768
	1.0	0.3	0.8113	0.7574	0.7785
		0.6	0.7831	0.6806	0.7267

## Data Availability

Not applicable.

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
