# Peer review of "Free Vibration Analysis of Functionally Graded Porous Cylindrical Panels Reinforced with Graphene Platelets"

_nanomaterials, 2023, doi:10.3390/nano13091441_

Round 1
Reviewer 1 Report
Review on the Manuscript:
Paper ID: nanomaterials-2334886
Title: Free Vibration Analysis of Functionally Graded Porous Cylindrical Panels reinforced with Graphene Platelets
I read the manuscript carefully. The free vibration of functionally graded porous cylindrical shell panel reinforced with graphene platelets is numerically investigated. In my opinion, the manuscript could be published after major corrections and a clearer and more careful wording:
1. The general rule is: after the numbered equations, put a comma if the following sentence starts with a lowercase letter, respectively put a point if the following sentence starts with a capital letter, see after equations (1) a point must be put, and after equation (2) a comma must be put and so on for the entire manuscript..
2. All new parameters and variables must be defined, for example in equation (2), the variables must be specified: h, z, |z| and V* CNT.
3. Likewise, WGPL in equation (3) is not defined.
4. In line 153, the porosity coefficient "e" is not defined.
5. Pages 6 and 7 present the theory of the finite element method, which can be summarized.
6. The upper indices of the variables in equation (31), (32) and line 228 are not specified.
7. The statement in lines 245 and 246 is not obvious.
8. In equation (36), the upper index "M" of the sum signum is not defined.
9. In equation (37) the matrix D is not defined.
10. In my opinion, the conclusions would be well numbered.
11. The author should to emphasize the novelty of the article and implicitly the improvements brought by this article in comparison with the realization from the literature.
03.04.2023
Author Response
Please refer to the response to reviewers' comments (1)

Reviewer 2 Report
Please see the attached file.

Author Response
Please refer to the response to reviewers' comments (2).

Reviewer 3 Report
The manuscript considers the free vibration of functionally graded porous cylindrical shell panel reinforced with graphene platelets. The effective material properties of the GPL-reinforced shell panel are evaluated by using the Halpin-Tsai model. Porosity distribution is considered. The cylindrical shell surface is transformed into the 2-D planar NEM grid.
The manuscript is organized with tables and diagrams. The following concerns must be considered before it can be accepted for publication.
1. Novelties of the paper should be better pointed out because it seems the numerical application of the 2-D planar natural element method.
2. English should be checked.
Author Response
Please refer to the response to reviewers' comments (3).

Round 2
Reviewer 1 Report
The authors considered all the recommendations made by the reviewers, for this reason, I recommend accepting the manuscript for publication. Anyway, the Editor-in-Chief has the final decision.